# Cytokine Profile in Predicting the Effectiveness of Advanced Therapy for Ulcerative Colitis: A Narrative Review

**DOI:** 10.3390/biomedicines12050952

**Published:** 2024-04-25

**Authors:** Hiroki Kurumi, Yoshihiro Yokoyama, Takehiro Hirano, Kotaro Akita, Yuki Hayashi, Tomoe Kazama, Hajime Isomoto, Hiroshi Nakase

**Affiliations:** 1Department of Gastroenterology and Hepatology, Sapporo Medical University School of Medicine, S-1, W-16, Chuo-ku, Sapporo 060-8543, Hokkaido, Japan; kurumi_1022_1107@yahoo.co.jp (H.K.);; 2Division of Gastroenterology and Nephrology, Department of Multidisciplinary Internal Medicine, Tottori University Faculty of Medicine, 36-1, Nishi-cho, Yonago 683-8504, Tottori, Japan

**Keywords:** biomarker, inflammatory bowel disease, molecular biomarker, cytokine profile, ulcerative colitis

## Abstract

Cytokine-targeted therapies have shown efficacy in treating patients with ulcerative colitis (UC), but responses to these advanced therapies can vary. This variability may be due to differences in cytokine profiles among patients with UC. While the etiology of UC is not fully understood, abnormalities of the cytokine profiles are deeply involved in its pathophysiology. Therefore, an approach focused on the cytokine profile of individual patients with UC is ideal. Recent studies have demonstrated that molecular analysis of cytokine profiles in UC can predict response to each advanced therapy. This narrative review summarizes the molecules involved in the efficacy of various advanced therapies for UC. Understanding these associations may be helpful in selecting optimal therapeutic agents.

## 1. Introduction

Ulcerative colitis (UC) is a chronic, remitting/relapsing inflammatory disease of the intestinal tract that requires lifelong monitoring and treatment [1]. Although the etiology of UC is not fully understood, abnormalities in the cytokine network contribute to the pathophysiology of UC. Several inflammatory bowel disease (IBD) risk loci are located in regions encoding cytokines and their downstream signaling mediators [2,3,4]. Therefore, advanced therapies such as cytokine-targeted therapies for UC have been developed and clinically used [5,6].

These advanced therapies include (1) anti-tumor necrosis factor (TNF)-α antibodies (infliximab, adalimumab, and golimumab), anti-interleukin (IL)-12/23p40 antibodies (ustekinumab), and anti-IL-23p19 antibodies (mirikizumab and guselkumab) that neutralize inflammatory cytokines; (2) Janus kinase (JAK) inhibitors (tofacitinib, filgotinib, and upadacitinib) that block downstream signaling of cytokine receptors; and (3) anti-integrins (vedolizumab and carotegrast) and sphingosine-1-phosphate (S1P) modulators (ozanimod) that inhibit migration of effector immune cells [7,8,9,10,11]. These therapies can directly or indirectly alter key inflammation pathways, receptors, and some crucial adhesion molecules [12]. They have shown efficacy in treating UC [13], but there are differences in how patients respond to each treatment. For instance, up to 40% of patients with UC do not respond to treatment with anti-TNF-α therapy [14]. This might reflect the fact that individual patients with UC have different cytokine profiles [15]. Several reports have explored the association between differences in cytokine profiles among individual patients with UC and therapeutic efficacy [16,17]. Multiple factors, including genetic background, environmental factors, luminal factors, and intestinal microbiota, are intricately involved in the etiology of UC, which could contribute to individual cytokine profile heterogeneity [18,19]. Therefore, it is reasonable that an approach focused on the cytokine profile of individual patients with UC would enhance the likelihood of success, minimize off-target effects, and produce sustained effects of cytokine-targeted therapy [20]. For instance, recent studies have demonstrated that Oncostatin M (OSM) is not only correlated with the severity of UC but can also predict the response to treatment with anti-TNF-α antibodies [21,22]. However, there is currently no evidence that individual cytokine profiles contribute to optimal biologics selection. 

This narrative review provides a summary from the perspective of cytokine profiles related to intestinal homeostasis and the pathophysiology of UC, and previous reports on cytokine profiles that predict therapeutic efficacy.

## 2. Role of Cytokines in the Pathophysiology of UC

### 2.1. Intestinal Homeostasis and Cytokinesis

Intestinal epithelial cells (IECs), immune cells, gut microbiota, and mesenchymal cells orchestrate the maintenance of intestinal homeostasis, which is regulated by cytokines. 

T helper (Th) 17 cells and type 3 innate lymphocytes (ILC3) maintain intestinal epithelial function by producing IL-6, IL-17, and IL-22. IL-6 and IL-22 promote IEC survival and proliferation through activation of signal transducer and activator of transcription (STAT) 3 signaling [23]. IL-17 promotes antimicrobial peptide secretion and IEC tight junction formation [24,25]. Gut microbiota stimulation triggers mononuclear phagocytes to produce IL-1β, IL-6, and IL-23, which in turn promotes IL-22 and IL-17 production by Th17 cells [26,27]. Dietary metabolites like retinoic acid (RA) and aryl hydrocarbon receptor (AHR) ligands also stimulate IL-22 production by ILC3 [28,29]. 

Type 1 interferon (IFN) produced by mononuclear phagocytes enhances barrier integrity through STAT1 and STAT2 signaling and promotes anti-inflammatory cytokine production by T regulatory (Treg) cells [30]. 

When IECs are damaged, macrophages are activated to promote the repair of the injured tissue. Classically activated M1 macrophages differentiate into M2 macrophages through STAT6 activation. M2 macrophages directly activate Wnt signaling pathway by expressing Wnt ligands, facilitating tissue repair [31,32]. 

Treg cells are essential for regulating excessive immune responses. IL-10 is a crucial anti-inflammatory cytokine secreted by Treg cells [33,34]. IL-10 inhibits both antigen presentation and subsequent release of pro-inflammatory cytokines. *IL-10* gene polymorphisms are associated with early-onset and severe intestinal inflammation [33]. Recently, the transcription factor cellular musculoaponeurotic fibrosarcoma (c-Maf) was shown to promote IL-10 production in forkhead box P3 (Foxp3)^+^ Treg and T regulatory type 1 (Tr1) cells [35]. Foxp3^+^ Treg cells express numerous IL-2 receptors, and IL-2 promotes Treg cell differentiation in the thymus and maintenance in the periphery [36]. IL-33 directly acts on Foxp3^+^ Treg cells, promoting their accumulation [37].

The intestinal epithelium is coated by secreted immunoglobulin (Ig) A to prevent pathogenic bacteria from interacting directly with the intestinal epithelium [38]. B cell activating factor (BAFF), IL-10, IL-6, and transforming growth factor (TGF)-β promote IgA secretion from plasma cells [39,40]. 

The interaction between mononuclear phagocytes, ILCs, T cells, and B cells is an important host adaptation to microbial stimuli, and its deficiency could be linked to the onset of IBD (Figure 1). 

### 2.2. Pathophysiology of UC

The immune system is divided into two main systems: innate and adaptive immunity. Innate immune cells respond rapidly and non-specifically to pathogens or other foreign entities. Once activated, they cause inflammation by releasing cytokines and chemokines and phagocytizing pathogens and cellular debris. Adaptive immunity takes longer to activate than innate immunity due to its dependence on antigen presentation and the cytokines produced by the innate immune response, but its response is specific. Both innate and adaptive immune cells have been shown to be significantly involved in the pathophysiology of UC [41].

#### 2.2.1. Innate Immunity

Disruption of the epithelial barrier function and microbial dysbiosis are important triggers in the pathophysiology of UC. This process involves the activation of the innate immune system by damage-associated molecular patterns (DAMPs) and pathogen-associated molecular patterns (PAMPs), which interact with pattern recognition receptors (PRRs) on IECs and immune cells. Immune cells such as neutrophils and mononuclear phagocytes and inflammatory cytokines like IL-1β, TNF-α, and IL-6 play crucial roles in the downstream of these responses.

Neutrophils are recruited from circulating blood to inflamed tissues by cytokines such as IL-1β, IL-6, and TNF-α and chemokines like C-X-C motif ligand (CXCL) 5 and IL-8 [42,43,44]. Abnormal activation of neutrophils has been observed in patients with UC [45]. Neutrophils directly promote tissue damage by releasing proteases such as matrix metalloproteinases (MMPs) and neutrophil elastases, and reactive oxygen species (ROS) [46]. The accumulation of activated neutrophils promotes changes in cryptic structures and the formation of crypt abscesses, mediated by an unbalanced enzymatic response, generation of TNF-α and IL-1β, and secretion of calprotectin [47,48,49]. 

In UC, lymphocyte antigen 6 complex (Ly6C)^high^ monocytes are increased at inflammatory sites. Their migration is regulated by the C-C motif chemokine receptor (CCR)2, IL-8 and TGF-β, which are constitutively produced by IECs [50,51,52]. Ly6C^high^ monocytes are highly susceptible to bacteria due to elevated expression of Toll-like receptor (TLR)2 and nod-like receptor (NOD)2 [53]. Additionally, C-X3-C motif chemokine receptor (CX3CR)^int^ macrophages with upregulated TLR2 also accumulate at inflammatory sites in UC [50,54]. These macrophages have an increased ability to secrete inflammatory cytokines such as TNF-α, IL-1β, and IL-6 [55]. Abnormally activated macrophages promote myofibroblast-mediated fibrosis by producing TGF-β1, connective tissue growth factor (CTGF), and fibroblast activating protein (FAP) [56]. 

Dendritic cells (DCs) link the innate and adaptive immune systems. Plasmacytoid DCs secrete type Ⅰ IFN, X-C motif chemokine receptor 1 (XCR1)^+^ myeloid DC1s are superior in antigen presentation to cytotoxic T cells, and signal regulatory protein α (SIPRα)^+^ myeloid DC2s polarize cluster of differentiation (CD)4^+^ T cell responses. Myeloid DCs are the dominant subtype in the intestinal lamina propria [57]. Activated DCs secrete inflammatory cytokines such as IL-1β, IL-18, IL-6, IL-12, and IL-23 to activate adaptive immunity. In UC, DCs exhibit an abnormal, immature phenotype with decreased expression of cutaneous lymphocyte antigen (CLA) and CCR4 and increased expression of CCR9 and β7 integrin [58,59,60]. Immature CD11c^+^CD11b^+^ myeloid DCs produce IL-23, which could sustain colitis [61]. The circulating cells in the peripheral blood of patients with active UC contain plasmacytoid DCs that migrate to secondary lymphoid organs, where they produce IL-6, IL-8, and TNF-α to perpetuate the disease [62]. 

ILCs are derived from common lymphoid progenitor (CLP) but are classified under the innate immune system. ILCs respond rapidly to signals and cytokines and act early in immune responses [63,64]. Retinoic acid receptor-related orphan receptor (ROR)γt^+^ ILC3 produces IL-17A and IFN-γ, promoting the pathogenesis of T cell-independent colitis [65]. Conversely, the frequency of natural cytotoxicity receptor (NCR)^+^ ILC3 producing IL-22 is reduced at inflammatory sites in patients with UC [66,67,68]. RORγt^−^ ILC1 promotes colitis through the production of IFN-γ and TNF-α [69,70]. ILC2 produces type 2 cytokines like IL-5 and IL-13, which contribute to epithelial barrier function and antigen clearance in the lumen but also play a role in chronic inflammation and tissue fibrosis [71].

#### 2.2.2. Adaptive Immunity

Under steady-state conditions, there are only a small number of CD4^+^ T cells in the intestinal epithelium. However, in patients with active UC, there is a significant increase in the number of CD4^+^ T cells in the intestinal epithelium [72]. Infiltration of T cells and accumulation of cytokines associated with T cells at inflammatory sites are common features of patients with UC [73,74,75]. Naïve CD4^+^ T cells are activated by antigen-presenting cells (APCs) in lymphoid tissues. Upon activation, these cells upregulate homing receptors, such as α4β7 integrins. Naïve CD4^+^ T cells differentiate into different subsets, including Th1, Th2, Th9, Th17, Th22, Treg, and Tr1 cells, depending on the microenvironment of various cytokines and transcription factors [72]. The cytokines produced by these subsets have multifaceted functions and complicatedly influence the pathophysiology of UC.

Several murine colitis models are characterized by excessive IL-12 and IL-23 production and decreased IL-10 and TGF-β production [76,77]. Both IL-12 and IL-23 are secreted by DCs and macrophages in response to early innate signals [78,79]. IL-12 induces Th1 cell differentiation and promotes IFN-γ and TNF-α production [80]. The identification of the *IL-23 receptor (IL-23R)* as a susceptibility locus for IBD and the clinical utility of anti-IL-23 antibodies confirm IL-23′s involvement in the pathophysiology of UC [81,82]. Prolonged IL-23 production in both humans and murine models converts barrier-promoting Th17 cells into pathogenic Th17 cells, leading to increased secretion of multiple cytokines, including IFN-γ, TNF-α, IL-17A, and granulocyte–macrophage colony-stimulating factor (GM-CSF) [83,84,85]. IL-23 also promotes intestinal inflammation by suppressing Foxp3^+^ Treg cells [86]. IL-1β and IL-23 collaborate to induce IL-17 production by Th17 cells and ILCs, promoting pathogenic responses [87]. The efficacy of anti-IL-12/23p40 antibodies against UC indicates that Th1 and Th17 cells are involved in the pathophysiology of UC [88].

On the other hand, Th2 and Th9 cells are abnormally activated at inflammatory sites in UC, leading to increased expression of their major cytokines, including IL-5, IL-13, and IL-9 [89,90]. Th2 cells are involved in extracellular microorganism elimination and support IgE-mediated B cell responses by secreting IL-4, IL-5, and IL-13, but their overactivation can contribute to chronic inflammation [72]. IL-5 is involved in the differentiation of eosinophils [91]. IL-33, a Th2 cytokine, is elevated at inflammatory sites in patients with UC compared to healthy controls [90,92,93,94]. Furthermore, blocking IL-33/suppression of tumorigenicity 2 (ST2) signaling has been shown to decrease disease activity, suggesting a pathogenic role for IL-33 [95]. 

IL-9 is a pro-inflammatory cytokine that activates JAK1 and JAK3 signaling upon binding to its receptor. Excessive IL-9 production in the intestinal tract can impair resistance to commensal bacteria by compromising the integrity of the epithelial barrier, leading to inflammation [96]. In addition, IL-9 promotes early population expansion of memory B cells and the production of IgG and IgE by B cells [97,98,99].

B cells can be classified into regulatory B cells secreting IL-10 and effector B cells secreting antibodies and various cytokines [100]. Regulatory B cells are decreased in the blood and intestinal tissues of patients with UC. Effector B cells cause inflammation by presenting antigens to T cells and secreting IL-2, IL-4, IFN-γ, TGF-β, and GM-CSF [101]. In contrast to the predominance of IgA in the intestinal mucosa of healthy individuals, IgG is predominant in the inflammatory mucosa of patients with UC [102,103]. The decreased IgA and increased IgG may be involved in the pathophysiology of UC [98]. IgG has a high affinity for antigens and activates the complement system. By binding to the fragment crystallizable (Fc)γ receptor, IgG promotes immune cell migration and maturation [101].

Indigenous bacterial-specific IgG antibodies are increased in the inflamed mucosa of patients with UC. In murine models, the induction of these antibodies induces intestinal inflammation by activating macrophage and Th17 cells and promoting neutrophil migration [104]. Fc receptor neonatal (FcRn) maintains IgG concentration by recycling IgG extracellularly. In a mouse model of dextran sodium sulfate (DSS)-induced colitis, specific inhibition of FcRn has been demonstrated to lower IgG concentrations and ameliorate colitis [105].

Both innate immune cells, including neutrophils, monocytes, macrophages, DCs, and ILCs, and adaptive immune cells, including T cells and B cells, are intricately involved in the pathophysiology of UC (Figure 2).

## 3. Mechanisms of Biologics and Small Molecule Compounds

### 3.1. Anti-TNF-α Antibodies

TNF-α is synthesized as a transmembrane TNF (mTNF), from which soluble TNF (sTNF) is released. sTNF binds preferentially to TNF receptor (TNFR) 1, while mTNF binds preferentially to TNFR2 [106]. TNFR1 is ubiquitously expressed and has a cell death domain, while TNFR2 is primarily expressed on lymphocytes and endothelial cells and lacks a cell death domain [107]. TNF-α primarily causes receptor-interacting serine/threonine-protein kinase (RIPK)1/3-dependent cell death via TNFR1 [108,109]. The TNFR2 pathway leads to the production of inflammatory cytokines such as IL-1β and IL-6, which have anti-apoptotic effects [110]. Anti-TNF-α antibodies inhibit TNFR1 signaling by neutralizing sTNF and TNF-α production by binding mTNF, which induces cell apoptosis through antibody-dependent cellular cytotoxicity, complement-dependent cytotoxicity, and outer-to-inner signaling. Anti-TNF-α antibodies also indirectly increase regulatory macrophages [111] and Foxp3^+^ Treg cells suppressing Th17 cells [112].

### 3.2. Anti-IL-12/23p40 Antibodies and Anti-IL-23p19 Antibodies

IL-12 and IL-23 are primarily produced by APCs. IL-12 acts on naïve T cells and promotes their differentiation into Th1 cells that produce TNF-α and IFN-γ [113]. IL-12 also induces the release of IFN-γ and TNF-α from CD8^+^ T cells, natural killer (NK) cells, and ILC1. Furthermore, IL-12 activates antimicrobial responses by DCs and macrophages [114]. IL-23 contributes to Th17 proliferation and stabilization and promotes the secretion of IL-17, TNF-α, and IL-22 by Th17 cells [115,116]. IL-23 also induces the secretion of IL-17 and IL-22 by γδ T cells and ILC3 [114] and the secretion of inflammatory cytokines such as TNF-α and IL-1β by binding to IL-23Rs on macrophages [117,118]. In addition, IL-23 suppresses IL-10 production from Treg cells, thereby reducing intestinal barrier and defense functions. Furthermore, experiments in mice have shown that IL-23 contributes to the differentiation, proliferation, and maintenance of Th2 cells [6]. 

Anti-IL-12/23p40 antibodies target p40, a common subunit of IL-12 and IL-23, and exert their anti-inflammatory effect by inhibiting both IL-12 and IL-23 signaling. Conversely, anti-IL-23p19 antibodies target p19, a subunit specific to IL-23, inhibiting only IL-23 without affecting other IL-12 family members [114].

Anti-IL-23p19 antibodies may be safer than anti-IL-12/23p40 antibodies due to the preservation of Th1 immune responses against infection and malignancies [119]. Although both anti-IL-12/23p40 and anti-IL-23p19 antibodies have been shown to alleviate experimental colitis to the same extent [25], there is no evidence suggesting a difference in the effects of these two antibodies in clinical practice. Basic research in mouse colitis models and clinical studies in patients with UC have shown that cytokine profiles differ depending on the disease stage of UC [6]. Th1- and Th17-cytokines are primarily involved in the pathogenesis of UC in the early phase, while Th2- and Th17-cytokines are primarily involved in the late phase. These differences could influence the choice of antibody treatment.

In addition, differences in binding affinity to target cytokines and specific effects via the Fc region of antibodies have also been reported. For example, guselkumab more effectively suppresses IL-23 secretion from CD64^+^ monocytes by binding to CD64 via the Fc region [120]. These differences in the molecular properties of antibodies may guide clinicians in selecting the most appropriate antibody for treatment.

### 3.3. Anti-α4β7 Integrin Antibodies

Anti-α4β7 integrin antibodies recognize a conformational epitope of the α4β7 integrin heterodimer. The transmembrane cell adhesion protein α4β7 is expressed on immune cells, including T cells, B cells, eosinophils, and ILC [121,122]. The ligand for α4β7, mucosal addressin cell adhesion molecule 1 (MAdCAM-1), is primarily found on endothelial cells within the gastrointestinal track and gut-associated lymph tissue [123,124]. Naïve T cells migrate to lymphoid tissues via high endothelial venules (HEVs) [125]. T cells activated by DCs or antigens in lymph nodes are transported from efferent lymphatic vessels through the S1P-S1P receptor (S1PR) signaling pathway [126]. Activated immune cells expressing α4β7 migrate from HEV-like vessels to intestinal tissues via the α4β7 and MAdCAM-1 signaling pathways [127,128]. Anti-α4β7 integrin antibodies suppress mucosal inflammation by inhibiting the migration of immune cells to the mucosal lamina propria.

### 3.4. JAK Inhibitors

Cytokines bind to specific receptors, causing activation and initiation of intracellular signaling pathways in various cell types [129]. JAK is a tyrosine kinase that binds to the intracellular domain of cytokine receptors [130]. When extracellular ligands bind to cytokine receptors, the JAK-STAT pathway is activated, and signals are transduced. The JAK family includes JAK1, JAK2, JAK3, and tyrosine kinase 2 (TYK2), with different combinations for each cytokine receptor [131,132,133]. In IBD field, currently available JAK inhibitors include tofacitinib, filgotinib, and upadacitinib, each with selectivity for different JAK families [5]. Compared to approaches that use antibodies to inhibit a single cytokine (e.g., TNF-α), JAK inhibitors have the potential to affect multiple cytokine-dependent pathways involved in the pathophysiology of UC (Table 1).

## 4. Molecules Predicting Therapeutic Efficacy

Previous reports on cytokine profiles predicting the therapeutic efficacy of each biologic in patients with UC are summarized in Table 2.

### 4.1. Anti-TNF-α Antibodies

Recent studies have demonstrated several mechanisms of resistance to treatment with anti-TNF-α antibodies. The mechanisms are briefly summarized below (Figure 3).

#### 4.1.1. Oncostatin M

OSM is a member of the IL-6 cytokine family. Oncostatin M receptor (OSMR) is expressed primarily on stromal cells, which produce IL-6, leukocyte adhesion factor, and chemokines in response to OSM [134]. 

West et al. reported that *OSM* mRNA expression is elevated in the inflamed mucosa of patients with UC who are resistant to anti-TNF-α antibodies compared to those who responded to anti-TNF-α antibodies [134]. Mucosal healing was achieved in 69–85% of patients with low *OSM* gene expression but only in 10–15% of patients with high expression. In addition, they reported that tissues with high *OSM* mRNA and *OSMR* mRNA expression showed enrichment in genes related to leukocyte chemotaxis, extracellular matrix organization, and mesenchymal development. Smillie et al. conducted single-cell RNA sequencing of colonic mucosa from patients with UC and healthy controls [141], revealing an increase in inflammation-associated fibroblasts (IAFs), a unique subset of fibroblasts in inflammatory tissues of patients with UC who are resistant to anti-TNF-α antibodies. The most enriched gene in IAF was *OSMR*, suggesting that downstream signaling activation of fibroblasts by OSM contributes to anti-TNF-α antibody resistance. Friedrich et al. also reported an increase in chemokine-high expressing fibroblasts in tissues from patients with IBD who were resistant to treatment with anti-TNF-α antibodies [142]. These results suggest that not only immune cells but also stromal cells, such as fibroblasts, may be involved in the therapeutic effect of anti-TNF-α antibodies [135].

The JAK-STAT pathway, activated by OSM, contributed to intestinal inflammation in a mouse model of DSS-induced colitis [143]. Clinical study in patients with IBD demonstrated a significant reduction in serum OSM levels in patients who respond to treatment with JAK inhibitors [144].

#### 4.1.2. IFN Signature

Type I IFNs, secreted by various cells, including DCs and fibroblasts, play an anti-inflammatory role by promoting epithelial repair and Treg cell differentiation. However, they also play a pro-inflammatory role by inducing IFN-stimulated gene, which has potent antimicrobial activity, and by promoting the production of inflammatory cytokines [145,146]. Type II IFNs, secreted by various cells including Th1 cells, ILC1, B cells, macrophages, and epithelial cells, provide broad protection against intracellular microbes [147]. Excess type II IFN exacerbates inflammation by activating macrophages, augmenting antigen processing, and inducing epithelial cell death [148].

*IFN* mRNA expressions in the blood and mucosa of patients with UC are heterogeneous [136,149]. High type I and type II IFN signatures in blood and colonic mucosa are associated with resistance to treatment with anti-TNF-α antibodies [136,150]. Mavragani et al. calculated type I IFN scores using *interferon-induced protein with tetratricopeptide repeat 1 (IFIT-1)* and *interferon-induced protein 44 (IFI-44)* mRNA expression. Similarly, type II IFN scores were calculated using *guanylate-binding protein 1 (GBP-1)* and *CXCL9* mRNA expression [136]. They reported that the group with lower baseline values for both type I and type II IFN scores had a better response to treatment with anti-TNF-α antibodies. Combining the type I and type II IFN scores predicted the response to treatment with anti-TNF-α antibodies with an area under the curve (AUC) of 0.98, sensitivity of 1.0, specificity of 0.88, positive predictive value (PPV) of 0.88, and negative predictive value (NPV) of 1.0.

Cell death signaling by TNF-α is mainly dependent on RIPK1/3 signaling [105,106], while IFN-induced cell death depends on JAK signaling [151]. It has also been suggested that the combination of TNF-α and IFN promotes cell death in colon epithelial cell lines through a TNFR1-independent synergistic effect [151,152]. It has been suggested that these differences in the mechanism of cell death may contribute to resistance to anti-TNF-α antibodies [152]. Flood et al. demonstrated that cell death induced by the cooperative effects of IFN-β and TNF-α is inhibited by JAK inhibitors using a colon organoid model [151]. Woznicki et al. showed that JAK inhibitors also inhibit IFN-γ-induced cell death using a colon cancer cell line [152]. 

#### 4.1.3. TREM-1

The triggering receptor expressed on myeloid cells 1 (TREM-1) belongs to the PRR family. TREM-1 is present on the surface of immune and non-immune cells, playing an important role in the host immune system [153]. TREM-1 recognizes PAMPs and DAMPs, collaborating with TLRs to amplify inflammatory innate immune responses [154].

Verstockt et al. reported that IBD patients with lower levels of *TREM-1* mRNA expression in their blood and mucosa were more likely to respond to treatment with anti-TNF antibodies and achieve endoscopic remission [137]. Czarnewski et al., using human homologs of differentially expressed genes identified by mRNA sequencing of the colon from mouse models of DSS-induced colitis, stratified patients with UC into two major transcriptome profiles (UC1 and UC2) [155]. Approximately 70% of UC2 patients responded to infliximab, compared to less than 10% of UC1 patients. Furthermore, approximately 60% of UC2 patients responded to vedolizumab (VED), while only 13% of UC1 patients did. TREM-1 was identified as the most accurate biomarker for classifying UC1 and UC2.

Prins et al. reported that the differentiation of monocytes into regulatory macrophages induced by anti-TNF-α antibodies was inhibited in monocytes expressing high levels of TREM-1 [156]. In addition, they reported a decrease in the expression of genes related to autophagy in these monocytes. Kökten et al. reported that inhibiting TREM-1 decreased the expression of mammalian target of rapamycin (mTOR) and increased the expression of autophagy related protein (ATG)1/UNC-51-like kinase (ULK)1 and ATG13 (which are involved in the initiation of autophagosome formation) and ATG5, ATG16L1, and microtubule-associated protein 1 light chain 3 (MAP1LC3)-I/II (which are involved in membrane elongation and expansion of formed autophagosomes) [157]. Because autophagy is indirectly involved in the differentiation and maintenance of regulatory macrophages [158,159], an increase in monocytes expressing high levels of TREM-1 may contribute to resistance to treatment with anti-TNF-α antibodies [160]. In addition, high TREM-1 expression may impair autophagy in intestinal epithelial cells, which promotes endoplasmic reticulum stress and exacerbates inflammation resistant to anti-TNF-α antibodies [157,161,162,163,164]. In a mouse model of DSS-induced colitis, pharmacological or genetic inhibition of TREM-1 has been shown to enhance macroautophagy and chaperone-mediated autophagy via mTOR dysregulation, thereby reducing endoplasmic reticulum stress and suppressing colitis [157].

#### 4.1.4. IL-23

Regarding a resistance factor to T cell apoptosis induced by anti-TNF-α antibodies, Schmitt et al. confirmed that IL-23 is upregulated in the mucosa of patients with IBD who do not respond to treatment with anti-TNF-α antibodies [165]. They found that a unique T cell subset, TNFR2^+^IL-23R^+^ T cells, was increased in the mucosa of these patients. TNFR2^+^IL-23R^+^ T cells were resistant to apoptosis induced by anti-TNF-α antibodies in an IL-23-dependent manner. They also reported that apoptosis-resistant TNFR2^+^IL23R^+^ T cells expressed integrin α4β7 and showed increased expression of IFN-γ, T-box protein expressed in T cells (T-bet), IL-17A, and RORγt compared to TNFR2^+^IL23R^−^ cells.

#### 4.1.5. IL-1β

Obraztsov et al. calculated Pearson correlation coefficients to analyze the relationship between 17 serum cytokine levels and the response to treatment with anti-TNF-α antibodies and developed a cytokine score using seven cytokine subsets including IL-1β, TNF-α, IL-12, IL-8, IL-2, IL-5, and IFN-γ [16]. In the study, patients received treatment with 5 mg/kg of anti-TNF-α antibody at weeks 0, 2, and 6, and their response to the treatment was evaluated at week 12. The score predicted clinical remission (Mayo score < 3) after three courses of anti-TNF-α antibody treatment with a sensitivity of 84.2%, specificity of 93.3%, and accuracy of 89.8% (44/49). Among several cytokine expressions, a high level of IL-1β is mostly associated with resistance to anti-TNF-α antibodies. Another report also suggests that high expression of IL-1β is associated with resistance to anti-TNF-α antibodies [16].

Bouwman et al. evaluated the association between activation of signaling pathways in the inflamed mucosa of patients with UC and responsiveness to treatment with anti-TNF-α antibodies [166]. They reported that the nuclear factor κ-light-chain-enhancer of activated B cells (NFκB), TGF-β, and JAK-STAT3 signaling pathways were activated in the mucosa of patients with UC who resistant to anti-TNF-α antibodies [166]. Candidate cytokines upstream of these signals were IL-1β and IL-17.

Friedrich et al. conducted RNA sequencing on surgically resected specimens from patients with IBD who did not respond to medical therapy [142]. They used weighted gene correlation network analysis to cluster co-expressed genes and identified 38 modules. Among these modules, two modules were associated with resistance to treatment with anti-TNF-α antibodies, and strongly linked to neutrophils and stromal cells. In addition, these modules were enriched with genes associated with inflammasomes. Furthermore, they identified inflammatory fibroblasts with high levels of chemokines that promote neutrophil recruitment using single-cell RNA sequencing. They reported that anti-TNF-α antibodies failed to suppress chemokine expression in these inflammatory fibroblasts, whereas IL-1 receptor inhibitors did. Thus, the activation of fibroblasts by IL-1β could contribute to resistance to treatment with anti-TNF-α antibody.

### 4.2. Anti-IL-12/23p40 Antibodies and Anti-IL-23p19 Antibodies

#### IL-22

In two randomized controlled trials, the administration of an anti-IL-23 antibody significantly reduced IL-22 levels in the blood and *IL-22* mRNA expression in the mucosa compared to the placebo group [81,167].

Pavlidis et al. assessed the IL-22 enrichment scores of colonic mucosa collected before ustekinumab initiation in patients with UC and observed substantial variation among patients [138]. Patients with low IL-22 enrichment scores had nearly twice the remission rates compared to all non-stratified patients, including clinical remission (25% vs. 13%) and mucosal healing (26% vs. 16%). Conversely, patients with high IL-22 enrichment scores showed outcomes similar to those in the placebo groups. The study confirmed that in patients resistant to anti-IL-12/23p40 antibodies, IL-22 expression depends on IL-1β rather than IL-23. Additionally, it demonstrated that IL-22 might induce the production of CXCL1 and CXCL5 from IECs via STAT3 activation and recruit C-X-C chemokine receptor 2 (CXCR2)-expressing neutrophils to the intestinal mucosa.

### 4.3. Anti-Integrin α4β7 Antibodies

#### 4.3.1. α4β7 Integrin

α4β7 integrin expression in immune cells contributes to the therapeutic effect of VED. Rath et al. reported that responders to treatment with VED had a higher mean number of α4β7^+^ immune cells per high power field in the intestinal lamina propria compared to non-responders (13.4 vs. 5.8, *p* = 0.0003) [168]. In a prospective clinical trial, responders to VED had lower baseline α4β7 expression of CD3^+^ and CD4^+^ T cells in peripheral blood compared to non-responders [169]. Furthermore, in responders, α4β7^+^ immune cells increased in peripheral blood and decreased in intestinal mucosa after VED administration [170], suggesting that the α4β7-MAdCAM-1 axis is the dominant recruitment pathway in VED responders.

Inflammatory cytokines like TNF-α promote MAdCAM-1 expression [171] and also increase vascular cell adhesion molecule 1 (VCAM-1), E-cadherin and L-selectin [172,173]. Baseline expression of α4β1 and αEβ7 in peripheral blood was not associated with the response to treatment with VED [169], but CD4^+^ T cells expressing αEβ7 and α4β1 in the peripheral blood increased after VED administration in non-responders [170,174]. Thus, redundancy in compensatory homing pathways may contribute to VED resistance. In a mouse model of enteritis unresponsive to α4β7 blockade, simultaneous inhibition of compensatory pathways such as L-selectin [175] and αEβ7 [176] has been shown to improve enteritis.

#### 4.3.2. IL-6 and IL-8

Bertani et al. reported that higher baseline serum IL-6 and IL-8 concentrations and decreased IL-6 and IL-8 after VED administration predict mucosal healing at week 54 [139]. However, the mechanism by which IL-6 and IL-8 predict the efficacy of treatment with VED is unknown. Several studies, including the above study, have shown that patients resistant to treatment with VED have higher disease activity, including longer disease duration, higher endoscopy scores, higher c-reactive protein (CRP), and lower albumin [140,177,178,179]. Elevated IL-6 and IL-8 levels may reflect disease severity. VED prevents the recruitment of new effector immune cells but has a limited effect on resident effector immune cells [180], making VED less effective in patients with severe disease activity. Combination therapy with other advanced therapies may overcome this limitation [181].

### 4.4. JAK Inhibitor

There are few reports on cytokine biomarkers that can predict the therapeutic effect of JAK inhibitors. While cytokines, such as serum IL-4 and IL-10, have been reported as potential biomarkers, their utility and underlying mechanisms have not been fully investigated [182]. Roblin et al. examined serum cytokine concentrations in patients with Crohn’s disease treated with filgotinib [144]. Although their study did not examine in detail whether baseline serum cytokine concentrations could predict treatment response, they found that significant reductions in IL-6 and OSM after treatment with filgotinib were significantly correlated with an endoscopic response (>50% reduction in simple endoscopic score for Crohn’s disease at week 10). As noted in the previous section, JAK inhibitors have been shown to alleviate inflammation induced by OSM and IFN in experimental mouse colitis models [143,151], suggesting the potential for JAK inhibitors to be effective in patients with UC exhibiting IL-6, OSM, and IFN-dominant cytokine profiles.

## 5. Prospects

Treatment options for UC have increased rapidly in recent years, leading to increasingly complex management algorithms, including which drugs to use, in what order, when to start, when to change doses, and when to discontinue. One of the obstacles to promoting personalized medicine for UC is the absence of a definitive biomarker for guiding treatment selection. The clinical application of cytokine profiles as a biomarker may be a feasible approach, because the underlying inflammatory cytokine profiles vary among patients with UC [183]. However, there is currently no evidence supporting the use of cytokine profiles to select optimal therapeutic agents. The complexity of differences in cytokine profiles between individuals, the variety of cytokine roles, and differences in racial and genetic backgrounds may make the use of cytokine profiles as biomarkers difficult [184].

This review summarizes the mechanisms underlying the therapeutic efficacy of advanced therapies for UC. Clinical trials in large cohorts are needed to determine the utility of cytokine profiles in selecting optimal therapeutic agents.

## Figures and Tables

**Figure 1 biomedicines-12-00952-f001:**
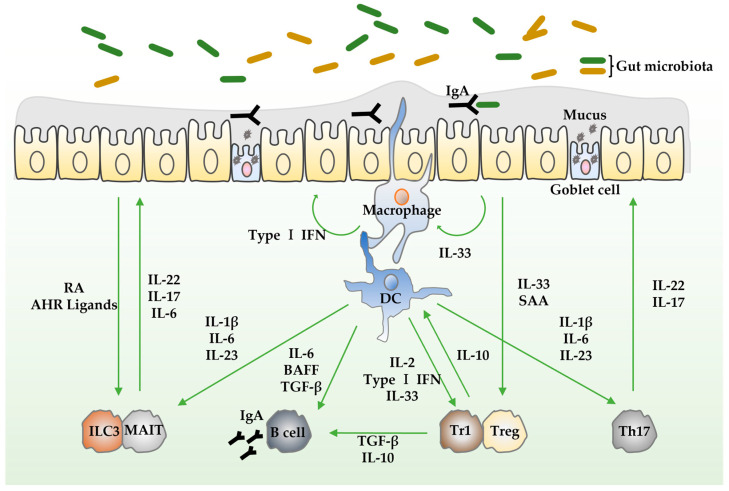
The role of cytokines that contribute to intestinal homeostasis. Intestinal homeostasis is maintained via interactions between IECs, immune cells, gut microbiota, and mesenchymal cells, and is tightly regulated by cytokines. Th17 cells and ILC3 promote IEC proliferation and survival, as well as antimicrobial substances and mucus secretion through the production of IL-22, IL-17, and IL-6. Mononuclear phagocytes enhance barrier integrity through the production of type I IFN. IL-10 produced by Tregs and Tr1 regulates excessive immune responses and maintains intestinal homeostasis. In addition, IgA secreted by B cells promotes effective defense by coating pathogenic bacteria from interacting directly with the epithelium. Abbreviations: AHR, aryl hydrocarbon receptor; BAFF, B cell activating factor; DC, dendric cell; IEC, intestinal epithelial cell; IFN, interferon; Ig, immunoglobulin; IL, interleukin; ILC, innate lymphoid cell; MAIT, mucosal-associated invariant T cells; RA, retinoic acid; SAA, serum amyloid A; TGF-β, transforming growth factor-β; Th, T helper; Tr1, Type 1 regulatory; Treg, T regulatory.

**Figure 2 biomedicines-12-00952-f002:**
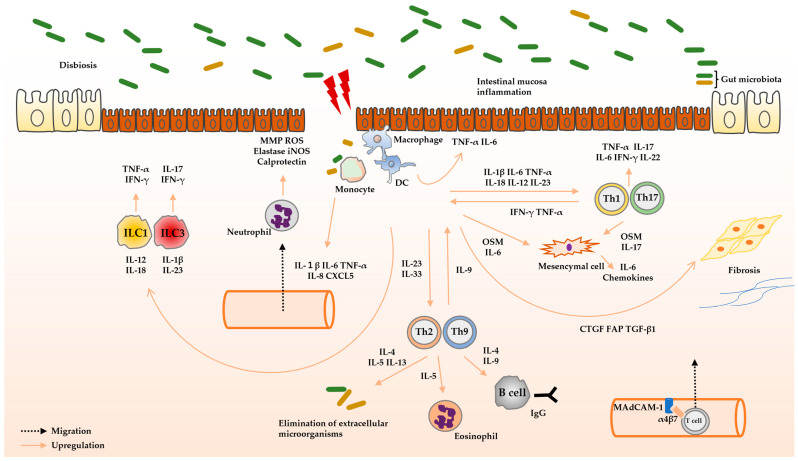
The role of cytokines that contribute to the pathophysiology of UC. The interaction of DAMPs and PAMPs with PRRs on the IECs and immune cells activates the innate immune system. Inflammatory cytokines such as IL-1β, TNF-α, and IL-6 play a central role in this process. These cytokines exert multifaceted effects on immune and non-immune cells, exacerbating the pathophysiology of UC. In UC, abnormally activated neutrophils release MMPs, ROS, elastases, iNOS, and calprotectin, which damage IECs. In UC, monocytes and macrophages with upregulated PRRs proliferate and have increased susceptibility to bacteria. ILCs promote the pathophysiology of UC via the release of IL-17, TNF-α, and IFN-γ. Naïve CD4^+^ T cells are activated by APCs in lymphoid tissues to upregulate homing receptors such as chemokine receptors and α4β7 integrin for T cell distribution to mucosa. Naïve CD4^+^ T cells differentiate into distinct subsets through the microenvironment of various cytokines and activation and repression of transcription factors, which contribute to the pathophysiology of UC. B cells are also involved in the pathophysiology of UC, as IgG is increased in the inflamed mucosa of patients with UC. Furthermore, OSM, IL-6, and IL-17 activate downstream signaling of mesenchymal cells, which promote inflammation. Abbreviations: APC, antigen-presenting cell; CD, cluster of differentiation; CTGF, connective tissue growth factor; CXCL, C-X-C motif ligand; DAMP, damage-associated molecular pattern; DC, dendric cell; FAP, fibroblast activating protein; IEC, intestinal epithelial cell; IFN, interferon; IL, interleukin; ILC, innate lymphoid cell; Ig, immunoglobulin; iNOS, inducible nitric oxide synthase; MAdCAM-1, mucosal addressin cell adhesion molecule 1; MMP, matrix metalloproteinase; OSM, oncostatin M; PAMP, pathogen-associated molecular pattern; PRR, pattern recognition receptor; ROS, reactive oxygen species; TGF-β, transforming growth factor-β; Th, T helper; TNF, anti-tumor necrosis factor; UC, ulcerative colitis.

**Figure 3 biomedicines-12-00952-f003:**
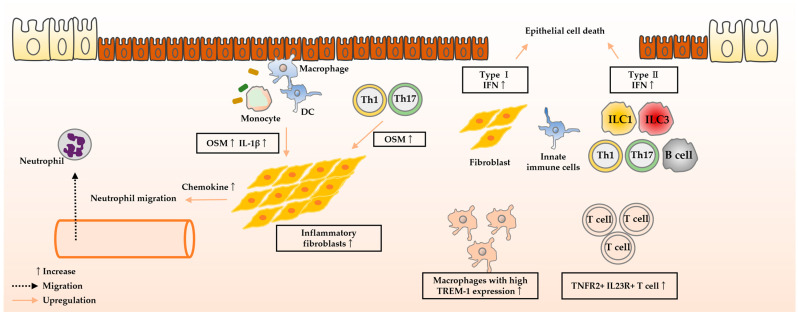
Factors contributing to resistance to treatment with anti-TNF-α antibodies. OSM and IL-1β induce chemokine production by inflammatory fibroblasts and promote neutrophil recruitment. This pathway may be resistant to treatment with anti-TNF-α antibodies. IFN induces epithelial cell death that is resistant to treatment with anti-TNF-α antibodies. Macrophages expressing high levels of TREM-1 are resistant to differentiation into the regulatory phenotype induced by anti-TNF-α antibodies. T cells expressing TNFR2 and IL-23R are resistant to apoptosis induced by anti-TNF-α antibodies. Abbreviation: DC, dendric cell; IFN, interferon; IL, interleukin; ILC, innate lymphoid cell; IL-23R, interleukin-23 receptor; OSM, oncostatin M; Th, T helper; TNF, anti-tumor necrosis factor; TNFR2, tumor necrosis factor receptor 2; TREM-1, triggering receptor expressed on myeloid cells 1.

**Table 1 biomedicines-12-00952-t001:** List of cytokines by JAK family.

	JAK1 and JAK3	JAK1 and JAK2	JAK1 and TYK2	JAK1 and JAK2, TYK2	JAK2 and TYK2	JAK2 and JAK2
Cytokine	IL-2, IL-4, IL-7, IL-9, IL-15, IL-21	IFN-γ	IFN-α, IFN-β, IL-22, IL-26, IL-10	IL-11, IL-13, IL-6, OSM, LIF	IL-12, IL-23, TPO	EPO, GH, IL-3, GM-CSF, IL-5
JAK inhibitor	Tofa, Filgo, Upa	Tofa, Filgo, Upa	Tofa, Filgo, Upa	Tofa, Filgo, Upa	(Tofa)	(Tofa)

Describes the relationship between cytokine receptors and the corresponding JAK family members. Abbreviations: EPO, erythropoietin; Filgo, filgotinib; GH, growth hormone; GM-CSF, granulocyte–macrophage colony-stimulating factor; IFN, interferon; IL, interleukin; JAK, Janus kinase; LIF, leukemia inhibitory factor; OSM, oncostatin M; Tofa, tofacitinib; TPO, thrombopoietin; TYK, tyrosine kinase; Upa, upadacitinib.

**Table 2 biomedicines-12-00952-t002:** Cytokine profiles and therapeutic efficacy of biologics.

	Molecule	Sample	Measurements	Outcome	Predicting Treatment Effect	Reference
Anti-TNF-α antibodies	OSM, OSMR	Mucosa	mRNA (qPCR)	Endoscopic and histologic remission	Mucosal healing (based on endoscopic and histologic criteria) was achieved in 69–85% of patients with low OSM module expression, but only 10–15% of patients with high OSM module expression.	[134]
	Panel (IL-13Rα2, TNFRSF11B, IL-11, STC1, PTGS2)	Mucosa	mRNA (qPCR)	Endoscopic and histologic remission	The panel divided responders and non-responders, with a sensitivity of 0.95 and specificity of 0.85.	[135]
	IFN	Blood	mRNA (qPCR)	Clinical and endoscopic remission and normalization of CRP	A low type I IFN signature score predicted response to anti-TNF-α antibody with an AUC of 0.95, sensitivity of 0.93, specificity of 0.88, PPV of 0.87, and NPV of 0.93. A low type II IFN signature score predicted response to anti-TNF-α antibodies with an AUC of 0.87, sensitivity of 0.86, specificity of 0.75, PPV of 0.75, and NPV of 0.86.	[136]
	TREM-1	Blood, Mucosa	mRNA (qPCR)	Endoscopic remission (MES ≤ 1)	Low whole blood and mucosal *TREM-1* mRNA levels predicted response to anti-TNF-α antibodies with AUCs of 0.78 (95% CI 0.65–0.90, *p* = 0.001) and 0.77 (95% CI 0.62–0.92, *p* = 0.003).	[137]
	Panel (TNF-α, IL-12, IL-8, IL-2, IL-5, IL-1β, IFN-γ)	Blood	Concentration	Endoscopic and histologic remission	The cytokine score had a sensitivity of 0.84, specificity of 0.93, and accuracy rate of 0.90 (44/49) for predicting response to anti-TNF-α antibodies.	[16]
Anti-IL-12/23 antibodies	IL-22	Mucosa	mRNA (qPCR)	Clinical remission (Mayo score of ≤2 and no subscore > 1) and Mucosal healing (Endoscopic and histologic remission)	Patients with low IL-22 enrichment scores had approximately twice as many clinical remissions (25% vs. 13%) and mucosal healing (26% vs. 16%) as all patients were not stratified.	[138]
Anti-α4β7 integrin antibodies	IL-6, IL-8	Blood	Concentration	Clinical remission (partial Mayo score of <2) and Endoscopic remission (MES of 0 or 1)	High serum IL-6 and IL-8 levels at baseline and decreased IL-6 and IL-8 levels 6 weeks after introduction of anti-α4β7 integrin antibodies predicted clinical remission with a sensitivity of 0.83 and specificity of 0.87, and endoscopic remission with a sensitivity of 0.82 and specificity of 0.90.	[139]
	IL-6	Blood	Concentration	Non-response (≤2 point decrease in Mayo score from baseline, 0 point decrease in rectal bleeding score or ≥1 point in rectal bleeding score)	High baseline serum IL-6 levels predicted resistance to vedolizumab with an AUC of 0.77 (95% CI: 0.57–0.98), sensitivity of 0.79, and specificity of 0.88.	[140]

Abbreviations: AUC, area under the curve; CI, confidence interval; CRP, C-reactive protein; IFN, interferon; IL, interleukin; MES, mayo endoscopic subscore; NPV, negative predictive value; OSM, oncostatin M; OSMR, oncostatin M receptor; PPV, positive predictive value; PTGS2, prostaglandin-endoperoxide synthase 2; qPCR, quantitative polymerase chain reaction; STC1, stanniocalcin 1; TNF, anti-tumor necrosis factor; TNFRSF11B, tumor necrosis factor receptor superfamily member 11B; TREM-1, triggering receptor expressed on myeloid cells 1.

## Data Availability

No new data were created or analyzed in this study. Data sharing is not applicable to this article.

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
