# Peer review of "Cytokine Profile in Predicting the Effectiveness of Advanced Therapy for Ulcerative Colitis: A Narrative Review"

_biomedicines, 2024, doi:10.3390/biomedicines12050952_

Round 1

Reviewer 1 Report

Comments and Suggestions for Authors

The manuscript by Kurumi et al. provides a narrative review on the“Cytokine Profile in Predicting the Effectiveness of Advanced Therapy for Ulcerative Colitis (UC).” It comprehensively summarizes the molecules involved in determining the efficacy of various advanced therapies for UC. The focus is sharp, and the content is of high quality, offering a valuable reference for understanding these associations, which may assist in the selection of optimal therapeutic agents. However, before accepting this manuscript, a few issues need to be addressed.

Abbreviations introduced in the article for the first time, such as TNF-α, require explanations. We request a thorough review of the entire text for the introduction and explanation of all abbreviations used.

It is advisable to relocate Figure 1 to the end of Section 2.1. Similar to the recommendation for Section 2.1, Section 2.2 should also include a detailed description rather than merely presenting a summary chart. Consequently, Figure 2 should be positioned at the end of Section 2.2.

There is a need for consistency in the formatting of Tables 1 and 2.

Author Response

Thank you very much for your suggestions. We have addressed your comments, and we feel that the manuscript has now been greatly improved as a result. Please see the revised manuscript and confirm our corrections.

Reviewer 1

â‘ Abbreviations introduced in the article for the first time, such as TNF-α, require explanations. We request a thorough review of the entire text for the introduction and explanation of all abbreviations used.

Thank you for your valuable comments. We have corrected the abbreviations in the manuscript.

â‘¡It is advisable to relocate Figure 1 to the end of Section 2.1. Similar to the recommendation for Section 2.1, Section 2.2 should also include a detailed description rather than merely presenting a summary chart. Consequently, Figure 2 should be positioned at the end of Section 2.2.

Thank you for your very valuable comments.

We have relocated Figure 1 and 2 to the end of their respective sections. We have also added the content to Section 2.2.

â‘¢There is a need for consistency in the formatting of Tables 1 and 2.

Thank you for your very valuable comments.

We have made the formatting of Table 1 and 2 consistent.

Thank you very much for your very valuable comments.

We have added the dosage of anti-TNF-α antibody in Paragraph 4.1.5.

We would like to thank the reviewer for their helpful comments. I hope that you will find the revised manuscript suitable for publication in Biomedicines.

Reviewer 2 Report

Comments and Suggestions for Authors

The paper is very interesting and fit for the publication but I have some observations:

- At page 7,line 2, the Authors speak about the role of Guselkumab that is not inserted in introduction

At page 10, paragraph 4.1.1, the Authors speak about the role of Oncostatin M and also this is not mentioned in introduction

At page 12, paragraph 4.1.5,  the Authors should explain the method used to dose anti TNF antibodies

Author Response

Thank you very much for your suggestions. We have addressed your comments, and we feel that the manuscript has now been greatly improved as a result. Please see the revised manuscript and confirm our corrections.

Reviewer 2

â‘ At page 7,line 2, the Authors speak about the role of Guselkumab that is not inserted in introduction.

Thank you very much for your very valuable comments.

We have added Guselkumab to the introduction.

â‘¡At page 10, paragraph 4.1.1, the Authors speak about the role of Oncostatin M and also this is not mentioned in introduction

Thank you very much for your very valuable comments.

We have added the content of Oncostatin M to the introduction.

â‘¢At page 12, paragraph 4.1.5,  the Authors should explain the method used to dose anti TNF antibodies

Thank you very much for your very valuable comments.

We have added the dosage of anti-TNF-α antibody in Paragraph 4.1.5.

We would like to thank the reviewer for their helpful comments. I hope that you will find the revised manuscript suitable for publication in Biomedicines.
